# Study of Windlass Mechanism in the Lower Limb Using Inertial Sensors

**DOI:** 10.3390/ijerph20043220

**Published:** 2023-02-12

**Authors:** María José Manfredi-Márquez, Sandra Priscila Tavara-Vidalón, Natalia Tavaruela-Carrión, María Ángeles Gómez Benítez, Lourdes María Fernandez-Seguín, Javier Ramos-Ortega

**Affiliations:** 1Podiatry Department, University of Seville, 41004 Sevilla, Spain; 2Physiotherapy Department, University of Seville, 41009 Sevilla, Spain; 3Institute of Biomedicine of Seville, 41013 Seville, Spain

**Keywords:** windlass mechanism, inertial sensors, radiographic angles, 1st metatarsophalangeal joint, lower-limbs kinematics

## Abstract

Aims: This study aimed to quantify the degrees of movement that occur in the lower limb using a kinematic system after taking two measurements of 45° and 60° of extension at the first metatarsophalangeal joint (1st MTPJ) and to test the validity of this sensor system using radiography. Methodology: This was a quasi-experimental test-post-test study with a single intervention group (25 subjects). Four inertial sensors were placed on the proximal phalange of the first toe, dorsum of the foot, medial-lateral of the leg (level of tibia), and medial-lateral of the thigh (level of femur). The extension of the 1st MTPJ produced movements of supination in the foot and rotation at the level of leg and thigh. We studied this mechanism in three situations (relaxed, 45°, and 60°) both with the sensors and with X-rays. Results: With the kinematic system, there was an increase in the range of movement in each of the variables, with a value of *p* < 0.05. The relationship between the kinematic system and the radiography was tested using Spearman’s rho test, obtaining a correlation coefficient of 0.624 and a value of *p* < 0.05, and the Bland–Altman graph, with 90% of the cases within the tolerance limits. Conclusions: The extension of the 1st MTPJ generated kinematic changes associated with supination movement in the midfoot and external rotation on the tibia and femur level. Both measurement techniques were very similar in the way that they quantified the degrees of extension of the 1st MTPJ. If we extrapolate this result to the measurement technique used by the inertial sensors, we could affirm that the values recorded in the supination and external rotation movements were reliable.

## 1. Introduction

The windlass mechanism (WM) is defined as the effect following the extension of the 1st metatarsophalangeal joint (1st MTPJ). This results in plantarflexion of the first radius and elevation of the internal longitudinal arch (ILA) through tension of the plantar fascia. Consequently, this causes supination of the subtalar joint (STJ) and an external rotation of the lower limb (LL), unaided by muscle action, all during the propulsive period of gait [1].

It is considered by authors such as Bolgla [2] to be a characteristic that is unique to human beings. It is essential in the development of bipedal gait and vital during the final phase of stance, converting the foot into a rigid lever with propulsive capacity. Fuller [1] described this mechanism as a mechanical model necessary to understand the movements and functionality of the foot. This author also detailed its development and how each of the anatomical structures involved in it act.

The alteration of any of these anatomical structures involved in its activation can cause its ineffectiveness during gait. This is associated with pathologies such as plantar fasciitis, hallux limitus, and hallux rigidus.

There are few studies that relate the WM with the movements that, during activation, may be generated in the LL. Most of them describe the bone and joint structures that constitute it and the movements that are being produced in the foot, making only a brief mention of the external rotation movement that occurs in the tibia and without addressing this in depth [3]. However, a quantification of the windlass mechanism for the lower limb does not exist. This mechanism is used as a maneuver during casting and the Jack test, and we think that it is necessary to know the degrees of rotation at the tibia and femur level.

The soft tissues are the main obstacle to recording bone and joint movements during movement analysis. This limitation was corroborated by different studies [4,5,6]. Peters et al. [4] conducted a systematic review where they quantified soft tissue artefacts during motion analysis in the LL and indicated that these depend on the location of the marker or sensor, the activity performed, the bone segment, and the anthropometric characteristics of the subjects participating in the analysis. Cereatti et al. [5], in agreement with McGinley et al. [6], stated that soft tissue artefacts represent one of the main obstacles during the performance of a kinematic study, making it difficult for the study to be accurate and reliable. Cereatti et al. [5] considered that these were difficult to reduce in a standardized way, as the methodology used, the subject selection criteria, the measurements being quantified, and the analysis systems are very varied. Even so, they tried to make a standardization proposal to minimize them.

In the biomechanical examination, we assessed the extension of the 1st MTPJ, we performed the Jack test, and we used the 1st MTPJ extension maneuver during casting. This maneuver is performed almost intuitively, as we do not know the relationship between the 1st MTPJ extension and the change in the lower limb. Therefore, our aim was to quantify the degrees of movement that occur in the LL by means of a kinematic system and through radiographs after performing two measurements of 45° and 60° of the extension in the 1st MTPJ.

## 2. Material and Method

### 2.1. Design

A quasi-experimental test-post-test study was conducted with a single intervention group and approved by the Ethics Committee of the Virgen Macarena and Virgen del Rocío University Hospitals of Seville (code number 0574-N-16), following the principles of the Helsinki Declaration.

### 2.2. Participants

The sample consisted of 25 subjects (17 women and 8 men) aged between 19 and 31 years. Healthy subjects were included if they presented a value of 0–5° valgus in relaxed calcaneal stance position (RCSP); foot posture index (FPI) with a value of 0–+5 [7]; ASA axis neutral position; and 60° extension of the 1st MTPJ [8]. Subjects were excluded if they had suffered fractures or surgery on the lower limb; had hallux abductus valgus or any inflammatory process or degenerative or rheumatic disease; had ligamentous hyperlaxity [9]; and/or were pregnant.

Data collection began at the Clinical Podiatry Area of the University of Seville after all participants signed the informed consent form.

### 2.3. Measurements and Study Protocol

Anthropometric and demographic data were obtained (age, sex, weight, and height). Subsequently, RCSP was calculated [10], and FPI was assessed as described by Redmond et al. [11]. Ligamentous laxity was assessed according to the Beighton scale [9], and the extension movement of the 1st MTPJ in unloading was quantified according to Munuera [8]. Finally, the ASA axis was located using the palpatory technique described by Kirby [12], and the plantar footprint was scanned with the representation of the ASA axis with the CbsScanFoot Digital Plantar Scanner model EDP-G2-A [13].

For the recording of the kinematic analysis, the Bioval Systems^®^ (Ticino, Switzerland) system was used, consisting of four inertial sensors placed on the proximal phalanx of the first toe (yellow), dorsum of the foot (red), medial level of the leg (green), and medial level of the thigh diaphysis (blue) of the right LL. This system allowed the quantification of the changes produced in the three planes of space in each of the sensors used. The green and blue sensor calculated the move of rotation of the lower limb at level of tibia and femur [14]. 

Three measurements of 30 s each were performed. The subject was in a standing position in a relaxed posture. The first toe was extended, and an EVA (ethylene-vinyl acetate) wedge of 45° extension was placed underneath it for about 10 s. The wedge was then removed, leaving the toe in a relaxed position. Next, the extension was performed again, this time with a 60° extension wedge for another 10 s. Finally, the toe was placed in a relaxed position. The movements recorded in degrees by the sensors were yellow sensor extension, red sensor prone/supination, and green and blue sensors rotation. For data analysis, the time sequence for each study situation (45° and 60°) was identified for the yellow sensor. Subsequently, the data were observed in the rest of the sensors in the same space of time, and the average of the three measurements was obtained.

Finally, three X-rays were taken of the loaded right foot, from a lateral projection, with the X-ray plate in contact with the medial side of the foot and the beam perpendicular to the plate at a distance of 1 m. All radiographs were taken with the source set at 53 kw and 5.0 mAs, and a Sedecal SPS HF-4.0^®^ (Madrid, Spain) and a portable X-ray unit with 24 × 30 cm Kodak X-Omatic^®^ collimator/cassettes (Rochester, NY, USA) were used. The first X-ray was taken with the foot relaxed and the second and third with a passive extension of the 1st MTPJ with an inclination of 45° and 60°, respectively. Subsequently, using AutoCAD^®^ software 2021, the metatarsophalangeal angle of the 1st toe (MTF 1°D) was quantified according to the measurement technique proposed by Palladino [15].

### 2.4. Statistical Analysis

The data were analyzed with SPSS software version 22.0 for Windows. Based on normality, for descriptive statistics, the mean and standard deviation for normal samples were used and the median and the 95% confidence index for non-normal samples. To study the evolution of the variables of the sensors, the Student’s *t*-test for related samples or the Wilcoxon test was used, and later, the effect size was calculated using the D_Cohen. To study the correlation between the sensor variables, the Pearson coefficient or Spearman’s rho correlation test was used. For the variable recorded by X-rays, an ANOVA of repeated measures and the Cohen’s d were used. Finally, to compare both systems, the Bland–Altman test was used. Statistical significance was considered for α = 0.05.

## 3. Results

A total of 25 participants with a mean age of 23.3 ± 3.3 years and BMI 22.5 ± 3.1 kg/m^2^ were recruited. The characteristics of the sample with respect to the variables RCSP grades, FPI, and 1st MTPJ grades measured with a goniometer and ASA axis location are shown in Table 1. Table 2 shows the mean, standard deviation (SD), and interquartile range (IC) for the variables 1st MTPJ, midfoot, tibia, and femur.

The radiological measurement of passive extension of the 1st MTPJ measured on X-ray was in relaxed load 10.4° (±1.7), in passive extension of 45° was 29° (±2.7), and passive extension of 60° was 39.4° (±3.8).

The behavior and evolution of the variables were analyzed in the different positions in which they were quantified with the inertial sensor system (45° and 60°), thus determining whether or not there were differences between them. The results (Table 3) indicated that there were statistically significant differences in the four variables studied. In addition, the magnitude of the difference between 45° and 60° was determined by assessing the effect size. It was observed that in both the 1st MTPJ and midfoot, the effect size had a low value of 0.200 and 0.202, respectively. Meanwhile, at tibia level (0.842) and femur level (0.686), the value was higher than 0.5, which is considered a large effect size in the tibia and medium in the femur. In short, the tibia and femur were the variables that presented the greatest difference in behavior between 45° and 60°. 

To determine whether or not there was a relationship and what type of relationship could be established between the independent variable, the 1st MTPJ, and the dependent variables, namely the midfoot, tibia, and femur, the parametric Pearson correlation test was performed for the midfoot and tibia, and the non-parametric Spearman’s rho correlation test was carried out for the femur. The results were not statistically significant (*p* > 0.05).

With regard to the radiographic measurement of the 1st MTPJ angle, the behavior and evolution of the variable were analyzed in the different positions in which it was quantified (relaxed load, 45° and 60°). The results indicated that there were statistically significant differences (Table 4). Post hoc tests were carried out, obtaining three pairs, i.e., CR-45°, CR-60°, and 45–60°, and the magnitude of the difference between 45° and 60° was determined, taking CR (relaxed load) as a reference, by assessing the effect size. The results obtained indicated that the CR-60° relationship was the most evident; i.e., it was the relationship where the greatest difference in behavior between the variables was observed. Regarding the magnitude of the difference between each of these three pairs, in the MTF 1°D variable, the effect size achieved had very high values: above 0.8. In the CR-45° pair, there was a value of 0.971, in CR-60° a value of 0.980, and in 45°–60° a value of 0.845.

Finally, the kinematic system was related to the radiography. For this, the variable 1st MTPJ quantified with the inertial sensor system and its radiographic equivalent, the MTF 1°toe angle, were used. First, Spearman’s rho correlation test was performed between the two variables. The result indicated that there was a direct relationship between them, with a correlation coefficient value of 0.624 and a value of *p* < 0.05, which was statistically significant. After obtaining this result, these two measurement techniques were compared based on both variables, using the Bland–Altman graphical method. The result indicated that the mean value was 13.3, the standard deviation was 5.5, the lower limit of tolerance was 2.52, and the upper limit of tolerance was 24.08. Ninety per cent of the cases were within the tolerance limits. In other words, similar results were obtained with both ways of measuring the extent of the 1st MTPJ.

## 4. Discussion

The aim of our study was to quantify the degrees of movement that would occur in the LL using a kinematic system after performing two measurements of 45° and 60° of extension in the 1st MTPJ. The objective was also to check the validity of these measurements by comparing this system with radiography. 

To test the validity of this system, radiography was chosen as a widely known, widely used, and validated technique for quantifying morphological and structural changes in the foot. Perlman et al. [16] studied and validated the protocol to be followed when performing a lateral radiograph of the loaded foot. 

Furthermore, they stated that the loaded lateral radiograph, respecting angle and base of support, is clinically similar to the findings observed during gait. The methodology for taking the radiographs in our research was based on this study. Along the same lines, Cavanagh et al. [17], following measurements, considered static structural variables to be significant predictors of foot function during gait. 

Bryant [18] compared a series of radiographic measurements taken under load with the feet in two different positions. The author’s study concluded by stating that there was a strong correlation between the values obtained by repeating the measurements in the same subject. Therefore, loaded radiography is considered a reliable method of foot assessment. 

As for the 1st MTPJ extension values that we established, they were also used by other authors. Cheng et al. [19] used the values of 15°, 30°, and 45° extension of the 1st MTPJ to analyze the behavior of the PF and Achilles tendon, relating the MW to the contraction force of the Achilles tendon. Caravaggi et al. [20] used wedges of 45°, 60°, and 75° extension under the 1st MTPJ. 

Regarding the results obtained with the sensor system, the extension of the 1st MTPJ triggers supination movements at the midfoot and external rotation at the tibia and femur. When comparing the results obtained between both measurements, we confirmed that there was an increase in the range of movement in each of the variables studied, with statistical significance (*p* < 0.05).

In the 1st MTPJ, the median value recorded in the first measurement was 42.5°, and the resulting value in the second was 52.6°, representing an increase in joint range of 10.1° between the two. These results indicate that, even when placing the finger on a surface of known angulation, in neither of the two measurements were the degrees (45° and 60°) reached, losing 2.5° and 7.4°, respectively. We interpret this to mean that when we perform a passive extension of the 1st toe, the PF exerts a certain resistance, causing an upwards force capable of raising the head of the 1st metatarsal. Therefore, the greater the extension, the greater the force. 

In the midfoot, the value obtained at 45° was 2.3° and at 60° was 3°, thus producing an increase of 0.7°. This is the dependent variable with the least representative result. It is possible that these values were a consequence of the blockage of the calcaneocuboid joint. Regarding LL rotations, the tibia recorded an increase of 1.3° and the femur an increase of 1.1° between each of the study situations (45° and 60°). This indicated that as the segment moved away from the intervention area, the effect decreased.

Of the four variables quantified, the tibia, with a value of *p* < 0.001, showed the greatest difference in behavior between the two measurements, with a large effect size of 0.842.

When the measurements were taken, the subject was in an anatomical position where the axis of the leg was located in the prolongation of the thigh axis. Seen in profile, the axis of the femur continues without any angulation with the leg. This reference position coincides with the extension of the knee joint, where the lower limb has its maximum length [21]. 

When the knee is in extension, the joint is much more stable [22]. Joint locking occurs, and the movement recorded by the sensors corresponds to the movement produced in the bony segments, accompanied by the soft tissues adjacent to them. This may explain why the movement recorded in the tibia was greater than that recorded in the femur.

Another justification could be that part of the mechanical energy we generated was transformed into thermal energy by the friction of the tissues themselves. The soft tissues are an artefact when it comes to quantifying the movement of the LL even though we used a device that minimized their effect on our results.

Therefore, we agree with Peters et al. [4] that the tibia is less susceptible to these alterations than the thigh. These authors, in their systematic review on artefact quantification, concluded that the thigh is the most vulnerable part of the LL, followed by the foot and ankle. Furthermore, they indicated that the latter would depend on the load applied to it. 

Of all the literature consulted, it is worth highlighting the contribution made by Caravaggi et al. [20,23], as their methodology was very similar to that of our research. Both studies used and compared a motion analysis system with lateral projection radiographs. In addition, they resorted to the use of 45° and 60° wedges to passively extend the 1st MTPJ in load and took as a reference in four of their models the position of the 1st MTPJ with respect to the ground, as we did with the modification of the metatarsophalangeal angle of the 1st toe. 

To find out and determine the degree of impact generated by the MW on the LL, we analyzed the effect size, the results of which, with a value of *p* < 0.001, were large for the tibia (0.842) and medium for the femur (0.686). 

We confirm, therefore, that the extension of the 1st MTPJ generated kinematic changes associated with midfoot supination and external rotation of the tibia and femur. 

The results obtained in the correlation coefficients between the variables determined that it was not possible to predict how many degrees of supination would be produced in the midfoot or the degrees of external rotation in the tibia or femur even when knowing the extension value of the 1st MTPJ. 

Finally, to answer our second objective, we compared the two measurement techniques used in the research. This was done by means of Spearman’s rho correlation test between the variables 1st MTPJ extension (inertial sensor system) and the MTF 1°D angle (radiography). The result was statistically significant, with a high correlation coefficient of 0.624 and a value of *p* < 0.05. A direct relationship was obtained between them; i.e., if one variable increases, the other increases as well. Based on the results obtained, we used the Bland–Altman graphical method. This indicated that 90% of the cases were within the tolerance limits. This can be translated as meaning that both techniques were very similar in their way of quantifying the grades of the 1st MTPJ. If we extrapolate this result to the measurement technique used by the inertial sensors, we could affirm that the values recorded in the supination and external rotation movements are reliable. 

## 5. Conclusions

The increase of the 1st MTPJ extension of 45° and 60° generates an increase of the kinematic movement of supination of the midfoot of 2.3 ± 1.1 and 2.9 ± 1.6, external rotation of the tibia of 4 ± 2.2 and 5.5 ± 3.1, and of the femur of 3.5 ± 1.6 and 4.4 ± 2.5, respectively, with a large effect size in the tibia and femur level. The measurement of the extension of the 1st MTPJ with inertial sensors and the measurement with radiography present a direct and significant relationship, so we can establish the similarity between both measurement techniques.

## Figures and Tables

**Table 1 ijerph-20-03220-t001:** Mean and SD of RCSP (valgus degree), 1st MTPJ, ASA (degree), and FPI.

	Mean	SD
RCSP	2.1	1.1
FPI	3.3	1.6
1st MTPJ	70.2	6.4
ASA	13.8	28

**Table 2 ijerph-20-03220-t002:** Descriptive analysis of the inertial sensor system according to joint extension (1^a^ MTPJ) at 45° or 60°. All variables are measures in degrees. 1^a^: first.

		Mean	SD	95 IC
1^a^ MTPJ	45°	42.6	3.7	39.7–44.8
	60°	52.6	4.6	50.1–55.4
Midfoot	45°	2.3	1.1	1.6–3.1
	60°	2.9	1.6	1.6–3.8
Leg (tibia level)	45°	4	2.2	2.2–5.6
	60°	5.5	3.1	2.6–8.5
Thigh (femur level)	45°	3.5	1.6	2.2–5.1
	60°	4.4	2.5	2.4–6.9

**Table 3 ijerph-20-03220-t003:** Analysis of the evolution of the variables in the inertial sensor system. ^1^ Student’s *t*-test for related samples. ^2^ Wilcoxon test. ** Large effect size. 1^a^: first.

	Mean ± SD (95 IC)		*p* (Effect Size)
	45°	60°	45–60°
1^a^ MTPJ	42.6 ± 3.7 (39.7–44.8)	52.6 ± 4.6 (50.1–55.4)	0.001 ^1^ (0.200)
Midfoot	2.3 ± 1.1 (1.6–3.1)	2.9 ± 1.6 (1.6–3.8)	0.005 ^1^ (0.202)
Leg (tibia level)	4 ± 2.2 (2.2–5.6)	5.5 ± 3.1 (2.6–8.5)	<0.001 ^2^ (0.842) **
Thigh (femur level)	3.5 ± 1.6 (2.2–5.1)	4.4 ± 2.5 (2.4–6.9)	0.001 ^2^ (0.686) **

**Table 4 ijerph-20-03220-t004:** Analysis of the evolution of the variable in the radiographic measurements. * ANOVA of repeated measures. ** Large effect size. 1^a^: first.

					*p* (Effect Size)		
	Relaxed Load	45°	60°	*p*	CR-45°	CR-60°	45°–60°
1^a^ MTPJ	10.4 ± 1.7 10 (9.0–12.0)	29 ± 2.7 29 (28.0–30.5)	39.4 ± 3.8 40 (37.5–41.5)	<0.001 *	<0.001 (0.971) **	<0.001 (0.980) **	0.002 (0.845) **

## Data Availability

The data associated with the paper are not publicly available but are available from the corresponding author upon reasonable request.

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
