# Peer review of "Study of Windlass Mechanism in the Lower Limb Using Inertial Sensors"

_ijerph, 2023, doi:10.3390/ijerph20043220_

Round 1

Reviewer 1 Report

The article is easy to read and logically structured. . This is a well written manuscript but some issues could be improved.-Shorten and specify key words.-In the introduction it would be interesting to include a clinically meaningful justification for the study.-Method: Why do you define a quasi-experimental design and not observational?- More precise detailing of sensor placement points

Author Response

Thank you for giving us the opportunity to resubmit our work. We have revised the manuscript according to your suggestions and comments. We have responded point by point to the comments and edited our manuscript accordingly. The changes are highlighted in yellow and listed in the order suggested by you. We thank you in advance for taking the time to review our manuscript. 

Reviewer 2 Report

The article examines a particularly important question related to the evaluation of the extension movement of the first metatarsophalangeal joint (1 MTPJ) and its influence on the kinematics of the lower limb. In clinical practice, the radiographic method is usually used to evaluate the extension of the 1st MTPJ. Of course, this is a fairly accurate method, but the human body is exposed to ionizing radiation. Therefore, the authors suggest the use of inertial sensors (IMU) as an alternative tool, which allows simultaneous evaluation of the kinematics of other anatomical structures. The results of inertial sensors are compared with the results of X-rays, thus the aim is to validate the data obtained by inertial sensors.

The research paper is interesting and valuable, but I have a few questions and comments to help improve it further:

1. First of all, I recommend that you think about the formulation of the title of the article, because the title of the article is a bit misleading. After all, the study examines not only extension of 1 MTPJ. In addition, the aim of the study is to quantify the movement of the lower limbs and the validation of the IMU sensor system. The RX abbreviation is also not entirely clear.

2. I suggest you consider changes to the abstract part: formulate one research objective; remove the inclusion criteria in the methodology section, and at the same time describe more the methodology of the research and the comparison methods used (inertial sensors measure the movements of the thigh, shank, foot, finger segments or hip, knee, ankle joints more than the tibia or femur movements); consider my further suggestions.

3. In my opinion, the introductory section should be strengthened by reviewing more literature related to the assessment of fine motor movements using IMU sensors. Consider moving the description of some sources from the discussion section to the introduction. I also suggest at the end of the introduction part to formulate one clear goal of the research and tasks to achieve that goal.

4. After reading the article, I was still unclear. Why are lower extremity kinematic parameters important? Why would they be useful in clinical practice?

5. I found the materials and methods section a bit odd. First, I would suggest dividing it into sub-sections "Participants", "Protocol and Data Acquisition", "Statistical Analysis" or similar. Inertial sensor attachment points (red, yellow, green and blue) and kinematic data (which segments/joints, which movements and which planes were evaluated) registration scheme would also be needed. Were both lower extremities assessed? Are participants segregated by gender or not?

6. In my opinion, some emphasis should be placed on the section of statistical analysis. What methods were used for which data according to their distribution? In order to confirm the validation, I would suggest defining certain parameter limits.

7. The results in Table 1 appear to be more than demographic. I would suggest to write the dimensions next to each presented parameter. What is the purpose of mean and median in tables? Typically, this depends on the distribution of the data and one of the following parameters is used.

8. The description of the results seems comprehensive, but at the same time confusing, as it is intertwined with the description of the statistical analysis methodology. I believe that a detailed description of the statistical analysis methods could update the description of the results.

9. In my opinion, the conclusions do not fully meet the purpose of the study.

Minor comments:

I suggest using the universally recognized term "relaxed calcaneal stance position (RCSP)" (line 16, line 69 and elsewhere).

Final phase of stance is pre-swing (line 43).

Look carefully at the abbreviations for some concepts, i.e., explain MI (line 51, 59 etc.), IPF (line 77 etc.), EVA (Ethylene-vinyl acetate) (line 89).

Anthropometric and demographic data were obtained… (line 76);

Wilcoxon (line 140)

What do the highlighted cells in Tables 3 and 4 mean?

Author Response

(The authors gave the same response as above.)

Round 2

Reviewer 2 Report

Thanks to the authors for taking into account most of my comments and suggestions, but I will make a few comments that bother me:

1.       Instead of the keywords "midfoot, tibia, femur", I would suggest simply adding "lower limbs kinematics".

2.       The title of Table 1 is incorrect. In Table 3 and Table 4, please note what the coloured cells mean (note under the table).

3.       The same rules for representing normally and non-normally distributed data apply everywhere (Table 2).

4.       I will repeat, usually, inertial sensors of systems like Bioval register the kinematics of the joints, taking into account the displacement of the segments on which they are placed. Therefore, it is incorrect to say that these are the bones of the tibia, femur movements (line 17, Table 2, Table 3 and etc). Also, the sensors are attached to certain segments and not to the bones. It is not clear to me what kind of things rotation is measured by the green and blue sensors (line 111), because no rotation movement is performed in the knee joint. I also still have uncertainties in the methodological part.

5.       My question: After reading the article, I was still unclear. Why are lower extremity kinematic parameters important? Why would they be useful in clinical practice? Your reply: Doesn´t exist a quantification of the windlass mechanism for the lower limb. This mechanism it used like maneuver during casting and jack test, we think that it is necessary to know the degrees of rotation in tibial and femur. I suggest putting this information in the article.

6.       The aim is to quantify the degrees of movement that occur in the LL by means of a kinematic system and through radiographs (lines 70-73). In conclusions, you state that the increase of the 1st MTJP extension affects LL movements. However, it is also stated that both measurement methods are similar. Is the goal to determine degrees of motion and validate the inertial sensor system? In my opinion, this does not serve the intended purpose. In addition, validation must be defined by certain accuracy criteria that are met or not met. Please review so that the purpose and conclusions are consistent throughout. I would also like to highlight what you are validating the system for? To avoid using X-rays?

Minor comments:

·         An error in the title of the article should be corrected “Mechanicsm” (line 2).

·         “tree” or “three” (line 17).

·         I would suggest adding the word "inertial sensors" (line 14 or line 18).

Author Response

Please find attached a document with the response to the reviewer. We welcome your suggestions and recommendations.
